# Stromal Vascular Fraction Cells from Individuals Who Have Previously Undergone Radiotherapy Retain Their Pro-Wound Healing Properties

**DOI:** 10.3390/jcm12052052

**Published:** 2023-03-04

**Authors:** Lucy V. Trevor, Kirsten Riches-Suman, Ajay L. Mahajan, M. Julie Thornton

**Affiliations:** 1Plastic Surgery and Burns Research Unit, University of Bradford, Bradford BD7 1DP, UK; 2Department of Plastic and Reconstructive Surgery, Bradford Teaching Hospitals NHS Foundation Trust, Bradford BD9 6RJ, UK; 3Centre for Skin Sciences, Faculty of Life Sciences, University of Bradford, Bradford BD7 1DP, UK

**Keywords:** radiotherapy, adipose-derived stem cells, stromal vascular fraction, dermal fibroblasts, wound healing, reconstructive medicine

## Abstract

Beneficial effects have been observed following the transplant of lipoaspirates containing adipose-derived stem cells into chronic wounds caused by oncologic radiotherapy. It is not yet certain whether adipose-derived stem cells are resistant to radiation exposure. Therefore, the aims of this study were to isolate stromal vascular fraction from human breast tissue exposed to radiotherapy and determine the presence of adipose-derived stem cells. Stromal vascular fraction from irradiated donor tissue was compared to commercially sourced pre-adipocytes. Immunocytochemistry was used to determine the presence of adipose-derived stem cell markers. Conditioned media from stromal vascular fraction isolated from irradiated donors was used as a treatment in a scratch wound assay of dermal fibroblasts also isolated from irradiated donors and compared to pre-adipocyte conditioned media and serum free control. This is the first report of human stromal vascular fraction being cultured from previously irradiated breast tissue. Stromal vascular fraction conditioned media from irradiated donors had a similar effect in increasing the migration of dermal fibroblasts from irradiated skin to pre-adipocyte conditioned media from healthy donors. Therefore, the ability of adipose-derived stem cells in the stromal vascular fraction to stimulate dermal fibroblasts in wound healing appears to be preserved following radiotherapy. This study demonstrates that stromal vascular fraction from irradiated patients is viable, functional and may have potential for regenerative medicine techniques following radiotherapy.

## 1. Introduction

Curative, adjuvant, or palliative radiotherapy is commonly used to combat cancer [1]. In breast cancer, a standard adjuvant radiotherapy regimen consists of 15 fractions of 40 Gγ external beam. Whilst the therapeutic aim is to provide enough radiation damage to halt or regress tumour growth, it is inevitable that adjacent healthy tissue is also exposed to radiation. Skin reactions are common, with over 90% of patients receiving radiotherapy reporting some dermal damage [2]. These reactions can be immediate—characterized by redness at the radiotherapy site—or prolonged in nature with dermal fibroblast (DF) dysfunction, hyper-inflammatory responses, fibrosis and damage to the underlying microcirculation [3].

Radiotherapy can be provided to breast cancer patients before or after mastectomy, and the impact of radiotherapy on success and satisfaction rates after subsequent reconstruction varies. For implant-based reconstruction, radiotherapy was associated with poor patient satisfaction irrespective of whether radiotherapy was applied prior to or after reconstruction. Similarly, prior, or post-reconstruction radiotherapy was associated with capsule contracture and an increased need for repeat surgery (reviewed in [4]). Interestingly, the degree of radiation-induced skin damage is negatively correlated with post-implant reconstructive complications [5]. For patients undergoing autologous tissue reconstruction with radiotherapy, patient satisfaction and quality of life are higher than for implant-based surgeries [6]. Given autologous implants use fat tissue from the stomach or back, it is possible that the improved outcomes in these patients are due to factors released from this adipose tissue.

Far from being an inert insulation system and connective tissue, adipose is now considered a complex endocrine organ. Approximately one third of adipose tissue is comprised of mature adipocyte cells, with the remaining two thirds comprising a heterogeneous mixture of cells including fibroblasts, endothelial progenitor cells, pericytes, white and red blood cells, as well as mesenchymal stem cells. These cells together comprise what is termed the stromal vascular fraction (SVF; reviewed in [7]).

The mesenchymal stem cell niche, termed adipose-derived stem or stromal cells (ASCs), were first identified in 2001 and were later isolated from human SVF [8,9]. Within the literature, the term ASC can be interchanged with preadipocytes [10]. They are present in SVF from white subcutaneous adipose tissue [11] and are reportedly more resistant to in vitro radiation exposure than bone marrow stem cells [12]. Thus, ASCs within human SVF could contribute to the observed beneficial impact of autologous breast reconstruction following radiotherapy. 

ASCs in culture are characterized by the expression of CD44, CD73, CD90 and CD105, with no expression of the hematopoietic marker CD45 (which is present in non-ASC SVF cells) [13,14]. ASCs secrete paracrine factors which can enhance wound healing in animal in vivo models [15] and ASC exosomes can promote DF migration in vitro [16]. The secretome contains multiple growth factors and adipokines including fibroblast growth factor 2 (FGF2), adiponectin and vascular endothelial growth factor (VEGF) [17,18,19]; all of which may promote the wound healing response.

Despite the promise that human ASCs hold for improved surgical outcomes following radiotherapy, and their reported resistance to radiation-induced damage, there have been no studies performed to date that have isolated ASCs from the SVF of patients who have undergone radiotherapy. Thus, the aim of this project was to perfect an isolation procedure of SVF from the breast tissue of patients who have undergone prior radiotherapy; to successfully culture ASCs from this SVF, and to provide proof-of-concept data that assesses whether their secretome may be beneficial for wound healing.

## 2. Materials and Methods

### 2.1. Human Tissue Collection

Human tissue was obtained via Ethical Tissue following ethical approval from Leeds (East) Research Ethics Committee. This was submitted 20 November 2017 with the ISAC reference number 17/091. Excess human tissue was obtained with informed consent from female patients undergoing reconstructive surgery following breast cancer, mastectomy/wide local excision and 40 Gy in 15 fractions of radiotherapy. Samples were transported on ice, at approximately 4–8 °C, and processed within 24 h.

### 2.2. SVF Isolation

Methodology was developed for the isolation of SVF cells from breast tissue subcutaneous fat adapted from the literature [20,21]. Subcutaneous fat was rinsed with tissue washing solution (sterile PBS with 10 µL/mL primacin), minced, and approx. 15 mL fat was transferred to a 50 mL tube containing 15 mL preadipocyte growth media with 1 mg/mL collagenase. This was briefly vortexed and incubated at 37 °C for one hour with regular (5–10 min) agitation. Following this, 15 mL preadipocyte growth media (PromoCell preadipocyte basal medium supplemented with 5% FBS, 4 µL/mL endothelial cell growth supplement, 10 ng/mL recombinant human epidermal growth factor, 1 µg/mL hydrocortisone, 90 µg/mL heparin and 2 µL/mL primacin) was added to the solution which was then passed through a cell strainer into a fresh 50 mL tube to remove undigested debris and centrifuged (600× *g* for 5 min). The floating fatty portion containing further undigested material and the supernatant was discarded and the pellet resuspended in 15 mL preadipocyte media. After further centrifugation (600× *g* for 5 min), the pellet was resuspended in 20 mL preadipocyte media and divided between two 75 cm^2^ flasks. After 24 h in a tissue culture incubator, adherent cells were washed thrice with PBS. Growth media was changed daily to ensure removal of red blood cells from the cultured cells. After 3–5 days the cells reached approximately 85% confluency and were passaged using trypsin/EDTA, centrifuged (600× *g* for 5 min) and re-seeded at a 1:3 dilution. 

### 2.3. Dermal Fibroblast (DF) Isolation

The epidermis and deep dermis/subcutaneous fat were separated from the dermis which was diced into 5 mm^2^ pieces. This was placed papillary dermis-down in a sterile 100 mm^2^ tissue culture dish in fibroblast growth medium (DMEM with 10% FBS, 2 µL/mL primacin and 10 µL/mL Glutamax). Tissue was incubated at 37 °C, in a humidified incubator with 5% CO_2_ in air and medium was changed after seven days and then twice weekly thereafter, allowing DF time to explant. DF were cultured in this way up until passage 3, at which point they were transferred into preadipocyte media. Cell growth and morphology were not altered. This media change ensured the same type of growth media was used throughout all experimental procedures, including conditioned media (CM) supplementation; and thus, any experimental differences in cell behaviour were attributable to factors secreted by cells rather than a different growth media composition. 

### 2.4. Preadipocyte Cell Culture

Commercial preadipocytes (PromoCell) were resuspended in preadipocyte media and transferred to 75 cm^2^ flasks and routinely passaged using trypsin/EDTA as above.

### 2.5. Immunocytochemistry

SVF cells and commercial preadipocytes were plated in 8-well chamber slides at a density of 5000 cells per well and grown in preadipocyte growth media until 70% confluent. They were then washed in PBS and fixed in ice-cold methanol for 10 min at −20 °C. Cells were washed thrice in PBS (5 min per wash), blocked in PBS containing 10% donkey serum (90 min) and incubated with primary antibodies diluted in PBS containing 1% donkey serum. Primary antibodies were: CD10 ab951 Anti-CD10 antibody (56C6) 1:100, CD45 ab40763 Anti-CD45 antibody (EP322Y) 1:100, CD73 ab54217 Anti-CD73 antibody (7G2) 1:200, and CD105 ab114052 Anti-CD105 antibody (3A9) 1:200. These were incubated for 1 h at room temperature. Cells were subsequently washed thrice in PBS (5 min per wash) and incubated with secondary antibodies diluted in PBS containing 1% donkey serum for one hour, protected from light. Secondary antibodies were: Alexa Fluor Donkey Anti-Mouse IgG and Alexa Fluor Donkey Anti-Rabbit IgG (versions for both that fluoresced at 594 or 488) 1:100. Finally, cells were washed four times with PBS and mounted using Vectashield^®^ Mounting Media containing 4′,6-diamidino-2-phenylindole (DAPI). Random images were captured on a confocal microscope at ×250 magnification.

### 2.6. Collection of Conditioned Media

SVF cells and commercial preadipocytes were plated into six-well plates at a density of 150,000 cells per well in preadipocyte media. After 48 h, cells were washed thrice in sterile PBS and maintained for 24 h in preadipocyte media containing no serum. Conditioned media (CM) was collected into sterile microcentrifuge tubes and centrifuged at 300× *g* for 5 min) before filtering through a 40 μm filter to remove cell debris. The resultant CM was stored at −20 °C until use.

### 2.7. Quantification of Paracrine Factors

The concentration of FGF2, adiponectin and VEGF in CM was measured using commercial Quantikine ELISA kits as per manufacturers’ instructions (R&D Systems). FGF2 was mixed 1:1 with assay diluent RD1W during the assay (DFB50); adiponectin was mixed 1:2 (DRP300) and VEGF was mixed 1:4 (DVE00). Unknown sample concentrations were interpolated from a standard curve run on each experiment.

### 2.8. Scratch Wound Assay

DF were seeded in triplicate at a density of 150,000 cells per well in six-well plates, in preadipocyte media. Once the cells reached 85% confluency they were washed thrice in sterile PBS and incubated for 24 h in preadipocyte media containing no serum. A linear wound was made in the monolayer, cell debris removed by washing with PBS, and the cell-free scratch wounded area imaged as previously described [22]. Media was replaced with serum free preadipocyte media, or SVF CM or preadipocyte CM as described in Section 2.6. A CooLpix 4500 Nikon camera was used to take images at 0, 6, 24 and 48 h at fixed points, at ×40 magnification.

The distance between the two wound edges (left and right) was measured at six fixed points per well using ImageJ software. The mean reduction in distance was calculated at each timepoint.

### 2.9. Statistical Analysis

Concentrations of paracrine factors were analysed using unpaired *t*-tests. Scratch wound migration assays used a two-way ANOVA with repeated measures and post-hoc test using GraphPad Prism 8. *p* < 0.05 was considered statistically significant. *n* refers to the number of independent experiments performed.

## 3. Results

### 3.1. Isolation of SVF

Isolation of SVF from irradiated tissue (IR-SVF) has not been performed previously. The methodology described here was adapted from protocols on non-irradiated SVF isolation as described in the literature [20,21]. During the optimisation of the technique, several amendments were made. Collagenase concentrations ranging from 0.5–2.0 mg/mL were tested for the release of IR-SVF from minced tissue; however 0.5 mg/dL failed to release cells from tissue and 2.0 mg/mL caused cell apoptosis thus a concentration of 1.0 mg/mL was used in the successful protocol. Red blood cells are prevalent in IR-SVF pellets and so two techniques were used to remove them from the resultant cell culture. In the first technique, the pellet was washed with an ammonium-chloride-potassium buffer to cause red blood cell lysis [21], however this also caused lysis of other cells within the IR-SVF and successful cultures were not explanted. Thus, we removed this lysis step and instead removed red blood cell contaminants by washing cultured cells with PBS daily to remove them. DF were explanted from tissue in parallel and the complete successful protocol for matched cell isolation can be seen in Figure 1.

### 3.2. Characterisation of Adherent Cells from SVF as Preadipocytes

The morphology of IR-SVF adherent cells was monitored across passages. At passage 0, there was an abundance of small, phase bright cells which were contaminating red blood cells; these were progressively removed during PBS washes as described previously. After 3–5 days, IR-SVF cells reached 85% confluence and were passaged. Subsequent to this, IR-SVF cells grew quickly and required passaging every two days. Morphology was initially heterogeneous at lower passages but became more uniform and became consistently spindle-shaped by passage 3 (Figure 2a). 

SVF can contain a heterogenous mixture of cells including stromal cells, pericytes, endothelial cells, red and white blood cells and ASCs/preadipocytes [7]. IR-SVF cells at passage 3 underwent immunocytochemical analysis to verify their identity, using a panel of preadipocyte markers. These were compared with commercially available human preadipocytes at passage 4 (Promocell). Due to issues with the supply of fluorescently conjugated secondary antibodies, IR-SVF immunocytochemistry used a 594 nm (red) tag and preadipocytes used a 488 nm (green tag); however, the primary antibodies used in both instances were the same (Figure 2b). 

The percentage of cells expressing these markers was calculated by assessing 100 random cells across multiple fields of view (identified by DAPI-stained nuclei) per antibody, per cell type. As expected, 100% of commercial preadipocytes stained positively for CD10, CD73 and CD105, and negatively for CD45. Only 58% of IR-SVF stained positively for CD10, whereas 93% were positive for CD73 and 100% were positive for CD105. All were negative for CD45 (Figure 2c). Cumulatively, this demonstrates that a high proportion of cells within the IR-SVF are ASCs/preadipocytes.

### 3.3. Conditioned Media from IR-SVF Promotes Wound Healing In Vitro

Preadipocytes are known to secrete numerous factors that can promote wound healing, including fibroblast growth factor 2 (FGF2), adiponectin and vascular endothelial growth factor (VEGF). Conditioned media (CM) from IR-SVF and commercial preadipocytes was collected and the concentration of these factors within CM was assessed using ELISA. The expression of both FGF2 and VEGF was comparable between preadipocytes and IR-SVF, however IR-SVF secreted significantly more adiponectin than preadipocytes (16.7 ± 0.45 ng/mL versus 14.0 ± 0.98 ng/mL); an increase of 20% (Figure 3a).

CM from IR-SVF and preadipocytes was used as a stimulus in DF migration assays to assess whether the secretion of paracrine factors by IR-SVF could still promote a wound healing response in vitro. CM from both IR-SVF and commercial preadipocytes promoted in vitro wound closure to a similar extent at all time points, with an increase in migration over control cells (treated with serum-free media only) visible as early as 6 h after treatment (Figure 3b,c).

## 4. Discussion

To our knowledge, this is the first example in the literature of obtaining and culturing viable cells from the SVF of patients who have been exposed to radiotherapy. A large proportion of cells within the SVF were ASCs (as demonstrated by expression of ASC/preadipocyte surface markers), and the secretome of these cells was able to promote in vitro migration of dermal fibroblasts. 

Our isolation protocol was adapted from previous methods established for non-irradiated adipose tissue [20,21] but had some key changes to improve success rates. Sample centrifugation speeds are notoriously variable between and sometimes even within individual laboratories, and in our case, we used a speed of 600× *g* for 5 min. Initial studies on the isolation of adipocytes from lipoaspirates suggested a higher speed of ~1200× *g* for 3 min [23]; however speeds upwards of 900× *g* have been proven to cause up to three-fold as much adipocyte cell death as lower speeds [24]; hence 600× *g* was a logical compromise between centrifugal stress and cell yield. Collagenase concentration was carefully titrated to balance sufficient tissue digestion with maintenance of cell membrane integrity; with optimum results observed at a concentration of 1.0 mg/mL. Furthermore, red blood cell lysis with ammonium chloride potassium buffer was omitted [21] from the protocol as we observed this also caused lysis of other cell populations within IR-SVF. Following these amendments, we were able to isolate adherent cells from IR-SVF which proliferated readily in culture and became more homogenous in appearance through routine subculturing.

Adherent cells from IR-SVF adapted an elongated spindle morphology typical of ASCs [25,26]. Analysis of cell surface markers demonstrated comparable positivity for CD73 and CD105 with commercial preadipocytes, as well as being CD45-negative. However, only half of IR-SVF cells were positive for CD10. Whilst CD10 has been found expressed in ASCs (reviewed in [27]), it is not a consensus as mesenchymal stem cells from human lipoaspirates were CD10 negative [28]. Thus, the absence of CD10 is not necessarily enough to conclude that ASCs only comprise up to half of the IR-SVF population. A further analysis of marker expression using Western blotting or flow cytometry would provide a more definitive characterization. 

Growth factors and chemokines are known to promote wound healing, and FGF2, adiponectin and VEGF have all had beneficial effects on in vitro wound healing assays [29,30,31]. We found that IR-SVF secreted each of these to a comparable level as commercial ASC/preadipocytes, and furthermore expressed 20% more adiponectin. Whilst adiponectin was originally thought to be secreted exclusively by adipose cells, it is now recognized that it can be secreted by alternative cell types including muscle and vascular cells [32]; it is likely therefore that the modest increase in adiponectin secretion by IR-SVF represents a contribution from non-adipose-restricted tissue. 

The secretion of these chemokines by IR-SVF suggested that CM from IR-SVF might be able to promote migration in a scratch wound assay. Conditioned media from ASC has previously been demonstrated to promote DF migration in a scratch assay [16]. IR-SVF also promoted DF migration in our scratch assay which lends further credit to our argument that the cells we have isolated from IR-SVF are ASCs and moreover, that these ASCs retain their functionality and are resistant to catastrophic damage caused by radiotherapy. The observation that IR-SVF and PA-CM were equally able to induce migration despite a significant difference in adiponectin expression suggests that adiponectin is not critical for DF migration. 

The immunomodulation, anti-inflammatory and proangiogenic activity of SVF and ASCs are similar, however, SVF is easier and quicker to obtain as it does not require isolation of different types of cells, and therefore in a clinical setting can be applied immediately. However, there are varying techniques (enzymatic vs. non-enzymatic) for SVF isolation, hence a more standardised and accepted protocol for application is required. While many clinical studies have reported the use of mesenchymal stromal cells (MSCs) to be safe and feasible, some minor side effects have been conveyed. Long-term cell culture can result in accumulation of abnormalities and use of antibiotics in culture media can increase risk of contamination with mycoplasma, although fibrosis is the most common adverse event described [33].

Nonetheless, the assessment of the effectiveness of cell monotherapy for wound healing and choice of cell sources is a promising biomedical approach. The complications resulting in poor healing in tissue that has been exposed to radiation damage is not dissimilar to other chronic non-healing wounds such as diabetic foot ulcers (DFU). To assess the regenerative effect of any cell type, it needs to be administered as a monotherapy in the absence of others with potential therapeutic effects. For example, application of somatic cells to DFUs only stimulates healing, while MSCs can contribute to the restoration of angiogenesis [34]. The therapeutic effects of various cells and clinical studies where stem and somatic cell-based therapy was administered as a monotherapy is reviewed in [34]. While studies have shown ASCs exhibit superior wound healing properties to other applications, e.g., hyaluronic acid, skin grafts, etc. reviewed in [35], their isolation is not an easy procedure and still requires consolidated clinical practice.

The culture of viable cells from SVF following exposure to radiation therapy is a novel finding. This valuable innovation is promising for the translation of ASC use into the clinic and reinforces the idea that mesenchymal stem cell populations may be resistant to radiation damage. This suggests that this tissue could still be a useful tool for promoting wound healing and surgical recovery in these patients. In vitro purification of these cells can be achieved in a matter of weeks, which again suggests a potential therapeutic application in patients, post-mastectomy. Further characterization of this cell population will clarify whether and how these cells have been affected by radiation exposure in vivo and shed light on this highly unexplored area of regenerative medicine. 

## Figures and Tables

**Figure 1 jcm-12-02052-f001:**
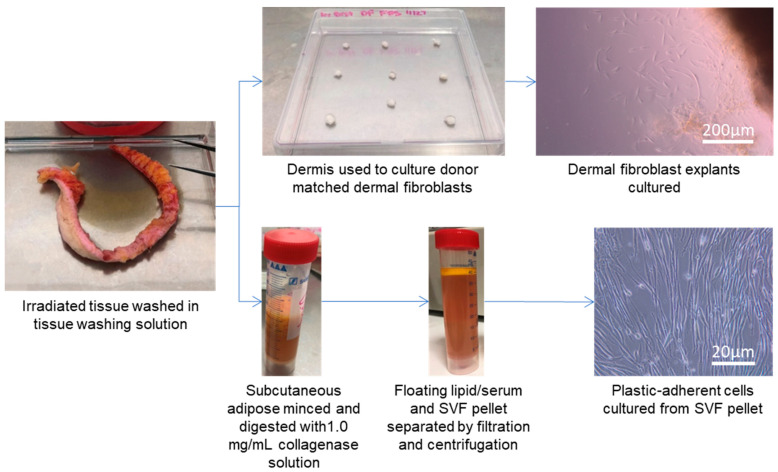
Optimised isolation of IR-SVF and DF from previously irradiated human breast tissue. Tissue was separated into two sections; one was used to isolate DF via an explant technique (dermal tissue can be seen as a dark region to the top right of the panel, with explanting cells radiating out of the tissue towards the bottom left corner). The second underwent mincing and digestion with 1.0 mg/mL collagenase, filtration, and centrifugation, before adherent cells from the IR-SVF were cultured. DF, dermal fibroblast; IR-SVF, stromal vascular fraction from irradiated tissue.

**Figure 2 jcm-12-02052-f002:**
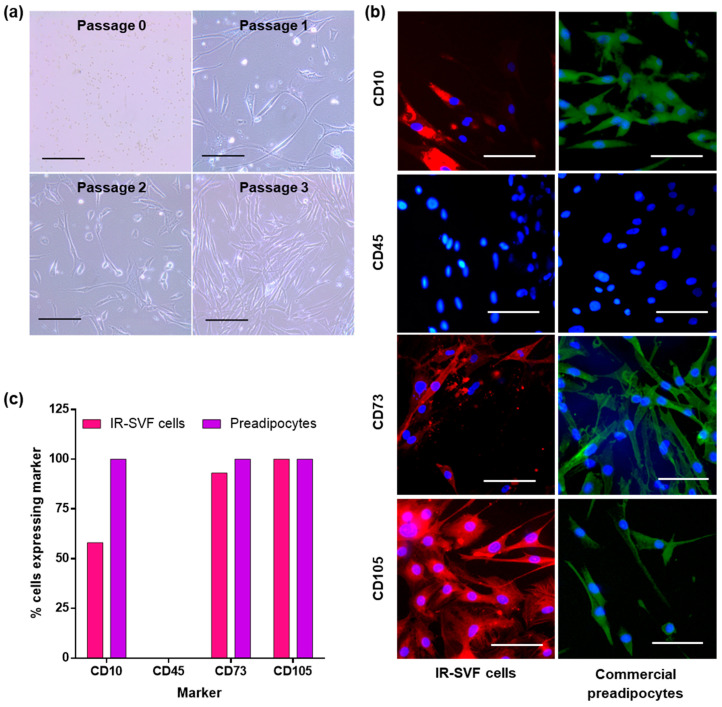
Characterisation of IR-SVF cells. (**a**) Random images of cells were taken using a light microscope at ×200 magnification, of IR-SVF from passages 0–3. (**b**) The expression of preadipocyte markers CD10, CD73 and CD105, as well as the negative control CD45, was assessed using immunocytochemistry in both IR-SVF and commercial preadipocytes. Multiple random fields of view were captured at ×250 magnification. Primary antibodies for each cell type were identical; secondary antibodies for IR-SVF were RFP-tagged and for preadipocytes were GFP-tagged. (**c**) From the immunocytochemistry images, the percentage of cells (identified by nuclei) staining positive for each marker was quantified and expressed as a percentage. *n* = 1 donor for each cell population; scale bars = 100 µm. GFP, green-fluorescent protein; IR-SVF, stromal vascular fraction from irradiated tissue; RFP, red-fluorescent protein.

**Figure 3 jcm-12-02052-f003:**
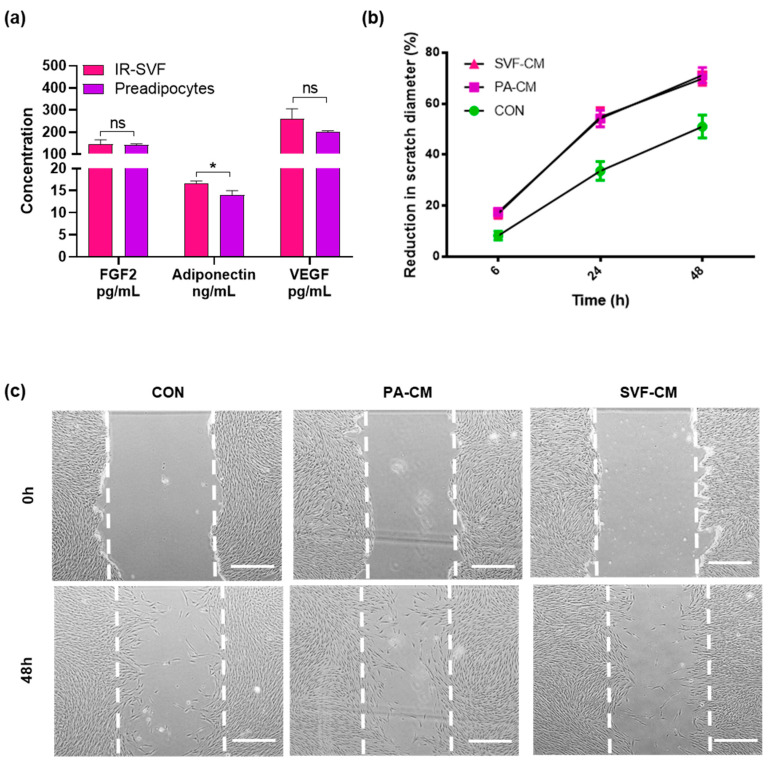
CM from IR-SVF promotes DF scratch wound healing to a comparable degree as CM from commercial preadipocytes. IR-SVF and commercial preadipocytes were cultured in serum-free media for 24 h and CM collected. (**a**) Expression of chemokines FGF2, adiponectin and VEGF was assessed using ELISA. (**b**) The ability of CM to promote in vitro wound healing of DF monolayers was monitored for up to 48 h. (**c**) Representative images captured at ×40 magnification, scale bars = 500 um, white dotted line is wound edge. *n* = 4; ns = not significant; * *p* < 0.05. CM, conditioned media; DF, dermal fibroblast; ELISA, enzyme-linked immunosorbent assay; FGF2, fibroblast growth factor 2; IR-SVF, stromal vascular fraction from irradiated tissue; VEGF, vascular endothelial growth factor.

## Data Availability

Data are available upon request.

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
