# Peer review of "Stromal Vascular Fraction Cells from Individuals Who Have Previously Undergone Radiotherapy Retain Their Pro-Wound Healing Properties"

_jcm, 2023, doi:10.3390/jcm12052052_

Round 1

Reviewer 1 Report

1. The aim of the presented study was to perform the stromal vascular fraction (SVF) isolation procedure from breast tissue of patients who have undergone prior radiotherapy; to successfully culture adipose-derived stromal cells (ASCs) from this SVF, and to provide proof-of-concept data that evaluate whether their secretome may be beneficial for wound healing.

2. Since the present study only evaluated the phenotypic characteristics of isolated cells, I recommend expanding the discussion by adding the advantages of SVF compared to other sources of Mesenchymal Stromal Cells. Among others, I recommend paying attention to adverse events, side effects and complications of cell therapy.

3. The assessment of the effectiveness of cell mono-therapy for wound healing and choice of cell sources is a promising biomedical approach. In fact, the obtaining and culturing viable cells from SVF of patients who have been exposed to radiation therapy is an extremely rare field of research. It should be noted that the same difficulties in healing these wounds are similar to some chronic disorders. I recommend a review of some research on the assessment of cell sources for chronic wound healing:

- Krasilnikova, O.A.; Baranovskii, D.S.; Lyundup, A.V.; Shegay, P.V.; Kaprin, A.D.; Klabukov, I.D. Stem and Somatic Cell Monotherapy for the Treatment of Diabetic Foot Ulcers: Review of Clinical Studies and Mechanisms of Action. Stem Cell Reviews and Reports 2022, 18, 1974–1985. https://doi.org/10.1007/s12015-022-10379-z

- Gentile, P.; Garcovich, S. Systematic Review: Adipose-Derived Mesenchymal Stem Cells, Platelet-Rich Plasma and Biomaterials as New Regenerative Strategies in Chronic Skin Wounds and Soft Tissue Defects. Int. J. Mol. Sci. 2021, 22, 1538. https://doi.org/10.3390/ijms22041538

4. In addition, I recommend expanding the 2020-2022 source citations in the Introduction and Discussion sections.

5. Lines 225-226. "The percentage of cells expressing these markers was calculated by assessing 100 cells (identified by DAPI-stained nuclei) per antibody, per cell type." - Did the authors really isolate exactly 100 cells in this study?

6. Line 275 etc.: '600G' - the acceleration of gravity is better denoted as small 'g'.

Reviewer 2 Report

The manuscript entitled “Stromal vascular fraction cells from individuals who have previously undergone radiotherapy retain their pro-wound healing properties” presents adipose-derived stem cells from stromal vascular fraction from human breast tissue exposed to radiotherapy were isolated and compared to commercially sourced pre-adipocytes. The ability of adipose-derived stem cells in the stromal vascular fraction to stimulate dermal fibroblasts in wound healing appears to be preserved following radiotherapy. Overall, the study is interesting. However, some issues needs to be addressed before publication. 

1. In Figure 1, the resolution of the optical microscopic image of isolated dermal fibroblast (DF) from the tissue should be improved seriously.

2. Statistical analysis should be performed for quantification of the percentage of positively stained cells for each marker in Figure 2 C.

3. Western bolting assays or Flow cytometry analysis is highly suggested for quantification of the percentage of positively stained cells for each biomarker, as immunofluorescence staining assay is only for Semi-quantification.

 4. Please explain why IR-SVF cells from irradiated tissue secreted significantly more adiponectin?

 5. As shown in Figure 3, IR-SVF secreted significantly more adiponectin than commercial preadipocytes, and the media from IR-SVF can successfully promote migration of DF cells, but showed no significant difference from commercial preadipocytes group. So what is the function of adiponectin in wound healing process?
